# Successive Years of Rice Straw Return Increased the Rice Yield and Soil Nutrients While Decreasing the Greenhouse Gas Intensity

**DOI:** 10.3390/plants13172446

**Published:** 2024-09-01

**Authors:** Meikang Wu, Min Nuo, Zixian Jiang, Ruiyao Xu, Hongcheng Zhang, Xiao Lu, Liqun Yao, Man Dou, Xu Xing, Xin Meng, Dongchao Wang, Xiaoshuang Wei, Ping Tian, Guan Wang, Zhihai Wu, Meiying Yang

**Affiliations:** 1Faculty of Agronomy, Jilin Agricultural University, Changchun 130118, China; wumeikang@mails.jlau.edu.cn (M.W.); 20220651@mails.jlau.edu.cn (M.N.); 20230789@mails.jlau.edu.cn (Z.J.); 15888293299@163.com (R.X.); 20230806@mails.jlau.edu.cn (H.Z.); 13848911109@163.com (X.L.); 15550272277@163.com (L.Y.); 13180633005@163.com (M.D.); xingxu@mails.jlau.edu.cn (X.X.); 15034815461@163.com (X.M.); dongchaowang@mails.jlau.edu.cn (D.W.); weixiaoshuang@jlau.edu.cn (X.W.); tianping@jlau.edu.cn (P.T.); wxkzbwg@163.com (G.W.); 2National Crop Variety Approval and Characterization Station, Jilin Agricultural University, Changchun 130118, China; 3Jilin Provincial Laboratory of Crop Germplasm Resources, Changchun 130118, China; 4College of Life Sciences, Jilin Agricultural University, Changchun 130118, China

**Keywords:** years of straw return, rice yield, greenhouse gas, soil nutrients

## Abstract

Straw return has important impacts on black soil protection, food security, and environmental protection. One year of straw return (S1) reduces rice yield and increases greenhouse gas (GHG) emissions. However, the effects of successive years of straw return on rice yield, soil nutrients, and GHG emissions in the northeast rice region are still unclear. Therefore, we conducted four successive years of straw return (S4) in a positional experiment to investigate the effects of different years of straw return on rice yield, soil nutrients, and GHG emissions in the northeast rice region. The experimental treatments included the following: no straw return (S0), a year of straw return (S1), two successive years of straw return (S2), three successive years of straw return (S3), and four successive years of straw return (S4). Compared with S1, the rice yields of S2, S3, and S4 increased by 10.89%, 15.46%, and 16.98%, respectively. But only S4 increased by 4.64% compared to S0, while other treatments were lower than S0. S4 increased panicles per m^2^ and spikelets per panicle by 9.34% and 8.93%, respectively, compared to S1. Panicles per m^2^ decreased by 8.06% at S4 compared to S0, while spikelets per panicle increased by 13.23%. Compared with S0, the soil organic carbon, total nitrogen, NH_4_^+^-N, NO_3_^−^-N, available phosphorus, and available potassium of S4 increased by 11.68%, 10.15%, 24.62%, 21.38%, 12.33%, and 13.35%, respectively. Successive years of rice straw return decreased GHG intensity (GHGI). Compared with S1, the GHGI of S4, S3, and S2 decreased by 16.2%, 11.84%, and 9.36%, respectively. Thus, S4 increased rice yield and soil nutrients, reducing GHGI.

## 1. Introduction

Rice is one of the most important types of grain food, accounting for 29% of global grain production since the 21st century [1,2]. China’s crop-straw production accounts for 25% of the world’s total production, and it has reached 800 million tons [3]. The amount of rice straw accounts for approximately 22.52% of the total crop-straw production in China. The rice planting area and total yield in the northeast rice region account for 18.33% and 19% of China’s total rice planting areas and yield, respectively [4]. The rice straw in the northeast rice region exceeds 20% of China’s total rice straw [4]. Therefore, the region of black soil in northeast China plays a crucial role in safeguarding the nation’s food security. However, due to the excessively intensive, unreasonable, or excessive use of cultivated land in this region, there has been a significant degradation of black soil [5]. This degradation not only reduces soil fertility, but also hampers the ecological service function, posing a threat to both national food security and regional ecological security [6]. As a renewable resource, straw is abundant in essential nutrients such as nitrogen, phosphorus, potassium, calcium, magnesium, and organic matter [7]. Straw return can increase soil organic carbon content; improve soil fertility, soil structure, and microbial characteristics; and promote the sustainable development of agriculture [8,9]. Consequently, conservation tillage practices (straw return) have been widely advocated in order to slow down soil degradation and achieve sustainable production of agricultural soils.

Straw return can affect the growth of rice. The presence of straw in the soil environment results in deterioration, leading to delayed revival and reduced tiller counts during the initial stage of rice growth [10]. However, the decomposition of straw releases nutrients that facilitate rice growth in the middle and later stages [11]. There are two main reasons why straw return inhibits the early tiller growth of rice. One reason is that the C/N ratio of rice straw is higher, which leads to competition for nutrients between microbes and plants, thereby affecting the growth of rice plants [12]. Another reason is that the decomposition of rice straw leads to the accumulation of microbial allelochemicals, such as organic acids, which restrict the development of rice seedling roots [13]. Nutrients and some small molecules released by straw decomposition in the middle and late stages of rice growth promote rice growth and yield formation. The availability of nitrogen uptake in rice plants has been found to be significantly influenced by the return of straw. This leads to a decrease in inorganic nitrogen content during the early vegetative stage, but also an increase in nitrogen supply during the panicle initiation stage. Consequently, there is an increase in the number of spikelets per panicle and a higher grain yield [14]. In addition, long-term return straw can delay leaf senescence; increase the rice leaf area index; improve the CO_2_ supply capacity of leaf pulp cells; increase the net photosynthetic rate, leaf SPAD value, and light energy interception rate of rice leaves; and promote grain filling, leading to increases in dry matter accumulation and yield [14,15]. In contrast, Zhang et al. (2021) found that, through a 9-year positional experiment under high fertilization levels, straw return does not have a significant impact on rice yield [16]. However, straw return did reduce the coefficient of variation of the rice yield by 25.8% and increased the sustainable yield index by 8.2%.

Straw return also increases GHG emissions from rice fields [17]. It has been found that CO_2_ emissions from straw return increase by 11.50–28.30%. This is mainly due to the fact that straw return can provide additional substrates for microbes and stimulate the decomposition of straw and native soil organic carbon content, thereby facilitating CO_2_ production [18]. Straw return has been shown to increase soil organic carbon content, resulting in the availability of more substrate sources for methanogens and ultimately promoting CH_4_ emissions [19]. It is noteworthy that straw return has significantly enhanced the CH_4_ emissions from paddy fields. Specifically, the increase in CH_4_ emissions following straw return is 4.3 times higher in Years 1–2, reaching its peak at 16.1 times higher in Years 6–7, and subsequently declining to 7.0 times higher in Years 12–13 [20]. This can be attributed to the fact that the degradation of soil organic carbon provides ample methanogenic substrates, thereby stimulating CH_4_ production and subsequent emissions [13]. Straw return increases the abundance of methanogens, rather than methanotrophs, in soil without a previous history of straw addition. This suggests that straw return has a stronger positive effect on CH_4_ production in the short term. However, in soil with a 12-year history of straw return, it significantly increases the abundance of both methanogens and methanotrophs, with a particularly higher increase observed in methanotrophs. This implies that straw return has a stronger positive effect on CH_4_ oxidation in the long term [21]. Straw return can indirectly affect soil nitrification and denitrification and, thus, soil N_2_O emissions by changing soil physicochemical and biological properties [22]. According to Abubakar et al. (2022), there exists a dynamic equilibrium between the fixation of mineral nitrogen and soil available nitrogen released from the decomposition of straw when the C/N ratio of the straw is within the range of 20 to 75 [23]. Additionally, the decomposition process of straw can deplete soil oxygen and promote anaerobic conditions, leading to increased denitrification. This, in turn, results in a significant conversion of N_2_O to N_2_ [24,25]. Consequently, almost no nitrogen is converted to N_2_O emissions into the atmosphere.

A large number of reports have focused on the amount of straw return; the method of straw return; the effects of fertilizers on soil fertility, soil microorganisms, and crop yields when applied to straw return; and other aspects [26,27,28]. However, there are fewer studies on the effects of different years of straw return on rice yield, soil nutrients, and GHG emissions in the rice fields of northeast China. Therefore, this experiment investigated the effects of different years of straw return on rice yield, soil nutrients, and GHG emissions through four successive years of straw return positional experiments, and it revealed the effects mechanisms through straw decomposition, root characteristics, and photosynthetic properties. Thus, it provides a theoretical basis and technical support for the rational utilization of straw, protection of black soil, improvement of crop yields, and realization of the sustainable development of agriculture in northeast China.

## 2. Results

### 2.1. Rice Yield, Dry Matter Accumulation, and Nitrogen Uptake of Rice

The years of straw return influenced the rice yield in both the year of 2021 and 2022 (Table 1). Compared with the S0 treatment, rice yield was, on average, decreased by 14.25%, 5.07%, and 1.13% (2021–2022) in the treatments of S1, S2, and S3, respectively. The S4 treatment increased the rice yield by 4.64% (2022) compared to the S0 treatment. Rice yield increased with increasing years of straw return. Compared with the S1 treatment, rice yield was significantly increased by 10.89%, 15.46% (2021–2022), and 16.98% (2022) in the S2, S3, and S4 treatments, respectively. Among the yield component traits, the main differences between treatments were observed in the spikelets per panicle and panicles per m^2^ (Table 1). Compared with S0, the panicles per m^2^ were, on average, decreased by 17.91%, 13.47%, 12.41% (2021–2022), and 8.06% (2022) in the treatments of S1, S2, S3, and S4, respectively. The panicles per m^2^ increased with increasing years of straw return. Compared with S1, the panicles per m^2^ were, on average, increased by 5.44%, 6.74% (2021–2022), and 9.34% (2022) in the treatments of S2, S3, and S4, respectively. Compared with S0, the spikelets per panicle were, on average, increased by 2.18%, 8.23%, 13.23% (2021–2022), and 16.14% (2022) in the treatments of S1, S2, S3, and S4, respectively. The rice harvest index of the different treatments showed the trend of S0 < S1 < S2 < S3 < S4 in both 2021 and 2022. Compared with S0, the rice harvest index was, on average, increased by 1.04%, 7.81%, 11.52% (2021–2022), and 13.44% (2022) in the treatments of S1, S2, S3, and S4, respectively.

Straw return increased the panicle length and panicle weight of rice (Figure 1). Compared with S0, the panicle length was, on average, increased by 3.06%, 5.84%, 8.78% (2021–2022), and 11.37% (2022) in the treatments of S1, S2, S3, and S4, respectively. In comparison to S0, the panicle weight was, on average, increased by 5.94%, 11.06%, 16.29% (2021–2022), and 16.62% (2022) in the treatments of S1, S2, S3, and S4, respectively. This showed that straw return can increase the panicle length and panicle weight of rice, and the trend increases with increasing years of straw return.

Years of straw return influenced the dry matter accumulation and nitrogen uptake of the rice in both years of 2021 and 2022 (Figure 2). At the TS, JS, HS, FS, and MS, the dry matter accumulation of S1, S2, S3, and S4 were lower than S0 and exhibited an order of S1 < S2 < S3 < S4 < S0 (Figure 2A,B). At the TS, the nitrogen uptake of the S1, S2, S3, and S4 treatments were lower than S0 and exhibited an order of S1 < S2 < S3 < S4 < S0 (Figure 2C,D). At the JS, HS, FS, and MS, the nitrogen uptake of the S1, S2, S3, and S4 treatments were lower than S0 and exhibited an order of S1 < S2 < S3 < S0 < S4 (Figure 2C,D). At the TS, the nitrogen uptake per plant of the S1, S2, S3, and S4 treatments were lower than S0 and exhibited an order of S1 < S2 < S3 < S4 < S0 (Figure 2E,F). At the JS, HS, FS, and MS, the nitrogen uptake per plant of the S1, S2, S3, and S4 treatments were lower than S0 and exhibited an order of S0 < S1 < S2 < S3 < S4 (Figure 2E,F). This indicated that the S1, S2, S3, and S4 treatments in the rice season reduced nitrogen uptake per plant at TS but enhanced the nitrogen uptake per plant of at the reproductive stage (Figure 2E,F).

### 2.2. Dynamics of Tiller Numbers, Leaf Area Index, and Net Photosynthetic Rate

Straw return significantly influenced the tiller number of the rice in 2021 and 2022 (Figure 3). The trends of the rice tiller changes were consistent across all treatments in 2021 and 2022, all showing an increasing and then decreasing trend with the reproductive process. The tiller number of rice in the S1, S2, S3, and S4 treatments were lower than S0 and exhibited an order of S1 < S2 < S3 < S4 < S0 (Figure 3A,B). Straw return significantly decreased the maximum tiller number of the rice in 2021 and 2022 (Figure 3C,D). Compared with the S0 treatment, the maximum tiller number of rice was, on average, decreased by 20.27%, 18.22%, 12.80 (2021–2022), and 7.00% (2022) in the treatments of S1, S2, S3, and S4, respectively. The maximum tiller number of the rice increased with increasing years of straw return. Compared with the S1 treatment, the maximum tiller number of the rice was, on average, increased by 18.22%, 12.80 (2021–2022), and 7.00% (2022) in the treatments of S2, S3, and S4, respectively. Straw return increased the percentage of productive panicles of the rice in 2021 and 2022 (Figure 3E,F). Compared with the S0 treatment, the percentage of productive panicles of the rice was, on average, increased by 3.03%, 4.89%, 7.10% (2021–2022), and 5.09% (2022) in the treatments of S1, S2, S3, and S4, respectively.

Years of straw return influenced the leaf area index and leaf area duration of the rice flag leaf in both years of 2021 and 2022 (Figure 4). At the TS and JS, the leaf area index of S1, S2, S3, and S4 were lower than S0 and exhibited an order S1 < S2 < S3 < S4 < S0 (Figure 4A,B). At the HS and FS, the leaf area index of the S1 and S2 treatments were lower than S0, while the leaf area index of the S3 and S4 treatments were higher than S0 and exhibited an order of S1 < S2 < S0 < S3 < S4 (Figure 4A,B).

At the TS-JS, the LAD of S1, S2, S3, and S4 were significantly lower than S0 and exhibited an order of S1 < S2 < S3 < S4 < S0 (Figure 4C,D). At the JS-HS and HS-FS, the leaf area duration of S1 and S2 were lower than S0, while the leaf area duration of S3 and S4 were higher than S0 and exhibited an order S1 < S2 < S0 < S3 < S4 (Figure 4C,D). This indicated that the leaf area index and leaf area duration of the rice flag leaf increased with increasing years of straw return.

Years of straw return influenced the net photosynthetic rate (Pn) and SPAD of the rice flag leaf in both years of 2021 and 2022 (Figure 5). The Pn and SPAD first increased and then decreased with the reproductive process, and it reached the maximum at HS. At the TS, the Pn and SPAD of the S1, S2, S3, and S4 treatments were lower than S0 and exhibited an order of S1 < S2 < S3 < S4 < S0 (Figure 5). At the JS, HS, FS, and MS, the Pn and SPAD of the S1 and S2 treatments were lower than S0 and exhibited an order of S1 < S2 < S0 < S3 < S4 (Figure 5). This indicates that the Pn and SPAD of rice flag leaf increases with increasing years of straw return.

### 2.3. Physiological Characterization of the Rice Root System

Straw return significantly influenced the activities of the enzymes related to nitrogen metabolism and root activity in the rice root system (Figure 6). The nitrate reductase (NR), glutamine synthase (GS), and glutamate synthase (GOGAT) of the root tended to increase with the reproductive process during the 40 days after transplanting. The NR, GS, and GOGAT of the root in the S1, S2, S3, and S4 treatments were significantly lower than S0 and exhibited an order of S1 < S2 < S3 < S4 < S0 (Figure 6A–C). The NR, GS, and GOGAT of the root increased with increasing years of straw return. The root activity first increased and then decreased with the reproductive process during the 40 days after transplanting. The root activity in the S1, S2, S3, and S4 treatments were significantly lower than S0 and exhibited an order of S1 < S2 < S3 < S4 < S0 (Figure 6D).

Straw return significantly influenced the root length and number of roots in the rice root system (Figure 7). The root length and number of roots tended to increase with the reproductive process during the 40 days after transplanting. The root length and number of roots in the S1, S2, S3, and S4 treatments were significantly lower than S0 and exhibited an order of S1 < S2 < S3 < S4 < S0 (Figure 7A,B). The root length and number of roots increased with increasing years of straw return.

### 2.4. Soil Nutrients

Straw return significantly influenced the soil chemical traits (Table 2). Compared with S0, the soil organic carbon content was, on average, increased by 3.14%, 6.42%, 10.22% (2021–2022), and 11.68% (2022) in the treatments of S1, S2, S3, and S4, respectively. Compared with S0, the soil total nitrogen content was, on average, increased by 2.85%, 5.92%, 9.31% (2021–2022), and 10.15% (2022) in the treatments of S1, S2, S3, and S4, respectively. There was no significant difference in the C/N ratio between treatments. Compared with S0, the available phosphorus content was, on average, increased by 2.99%, 6.96%, 10.78% (2021–2022), and 12.33% (2022) in the treatments of S1, S2, S3, and S4, respectively. Compared with S0, the available potassium content was, on average, increased by 5.76%, 8.63%, 11.02% (2021–2022), and 13.35% (2022) in the treatments of S1, S2, S3, and S4, respectively.

Years of straw return influenced the NO_3_^−^-N and NH_4_^+^-N contents in both years of 2021 and 2022 (Figure 8). The NO_3_^−^-N and NH_4_^+^-N contents decreased with the reproductive process. At the TS, the NO_3_^−^-N and NH_4_^+^-N contents of S1, S2, S3, and S4 were lower than S0 and exhibited an order of S1 < S2 < S3 < S4 < S0 (Figure 8). At the JS, HS, FS, and MS, the NO_3_^−^-N and NH_4_^+^-N contents of S1, S2, S3, and S4 were higher than S0 and exhibited an order of S0 < S1 < S2 < S3 < S4 (Figure 8). This indicated that the NO3--N and NH4+ contents increased with increasing years of straw return.

### 2.5. Nutrient Release of Rice Straw

Years of straw return influenced the decomposition of straw in 2022 (Figure 9). The remaining dry matter, carbon, and nitrogen of the straw decreased with the reproductive process during the rice growing season. The remaining dry matter, carbon, and nitrogen of straw in the S2, S3, and S4 treatments were lower than S1 and exhibited an order of S1 < S2 < S3 < S4 (Figure 9). The remaining dry matter, carbon, and nitrogen of straw decreased with increasing years of straw return. The straw decomposition rate, percentage of carbon release, and percentage of nitrogen release increased with reproductive process during the rice growing season. The straw decomposition rate, percentage of carbon release, and percentage of nitrogen release in the S2, S3, and S4 treatments were higher than S1 and exhibited an order of S1 < S2 < S3 < S4 (Figure 9). The straw decomposition rate, percentage of carbon release, and percentage of nitrogen release increased with increasing years of straw return (Figure 10).

### 2.6. CO_2_, CH_4_, and N_2_O Emissions

The seasonal variations in the CO_2_ flux followed similar patterns, regardless of treatments (Figure 11A,B). The CO_2_ flux always gradually increased from the beginning of the seasons and then peaked at the mid-season drainage. The CO_2_ emissions during the rice season were mainly concentrated from the transplanting stage to the panicle fertilizer (Table 3). Compared with S0, the CO_2_ emission was, on average, decreased by 19.65%, 34.61%, 28.70% (2021–2022), and 24.55% (2022) in the treatments of S1, S2, S3, and S4, respectively. No significant difference was found in the CO_2_ emissions in the S1, S2, S3, and S4 treatments in all rice growing seasons.

The seasonal variations in the CH_4_ flux followed similar patterns, regardless of treatments (Figure 11C,D). The CH_4_ flux always gradually increased from the beginning of the seasons and then peaked at the period before mid-season drainage. Mid-season drainage sharply depressed the CH_4_ flux from each treatment. Along with reflooding after mid-season drainage, the CH_4_ flux slightly increased and peaked again in the middle–late season, and it then kept at a very low value until harvesting. The CH_4_ emissions during the rice season were mainly concentrated from the transplanting stage to the mid-season drainage, with relatively less CH_4_ emissions from the mid-season drainage to the final mature harvest (Table 3). Compared with S0, the CH_4_ emission was, on average, decreased by 24.7%, 28.35%, 23.33% (2021–2022), and 19.03% (2022) in the treatments of S1, S2, S3, and S4, respectively. No significant difference was found in the CH_4_ emissions in the S1, S2, S3, and S4 treatments in all rice growing seasons.

The seasonal variations in N_2_O flux followed similar patterns, regardless of treatments (Figure 11E,F). The peak of N_2_O flux occurred after the application of tillering fertilizer, panicle fertilizer, and mid-season drainage. The N_2_O emissions during the rice season were mainly concentrated from the transplanting stage to after the mid-season drainage, with relatively less N_2_O emissions from after the mid-season drainage to the final mature harvest (Table 3). No significant difference was found in the N_2_O emissions in the S0, S1, S2, S3, and S4 treatments in all rice growing seasons.

### 2.7. The GWP and GHGI

Straw return significantly increased the GWP and GHGI (Figure 12). Compared with S0, the GWP was, on average, increased by 22.63%, 30.94%, 26.97% (2021–2022), and 21.34% (2022) in the treatments of S1, S2, S3, and S4, respectively. No significant difference was found in the GWP in the S1, S2, S3, and S4 treatments. Compared with S0, the GHGI was, on average, increased by 43.57%, 38.21%, 27.98% (2021–2022), and 15.94% (2022) in the treatments of S1, S2, S3, and S4, respectively. The GHGI decreased with increasing years of straw return. Compared with S1, the GHGI was, on average, decreased by 6.02%, 11.72% (2021–2022), and 14.21% (2022) in the treatments of S2, S3, and S4, respectively. 

### 2.8. Correlation Analysis

Correlation analysis showed that the RY was highly significantly and positively correlated with the SP, DMA, NU, LAI, Pn, NR, GOGAT, GS, RL, and Nr (Figure 13). The RY was significantly and positively correlated with the PM, PL, MTN, PPS, TN, NO_3_^−^-N, NH_4_^+^-N, SDR, and PCR. The RY was significantly and negatively correlated with the GHGI. CO2 was highly significantly and positively correlated with the PL, PW, NCP, Pn, TN, NH_4_^+^-N, AP, and AK. CO_2_ was significantly and positively correlated with the SP, PPS, SPAD, SOC, SDR, PCR, and PNR. CO_2_ was significantly and negatively correlated with the MTN and PM. CH_4_ was highly significantly and positively correlated with the PL, PW, NCP, PPS, Pn, TN, NH_4_^+^-N, AP, AK, SDR, PCR, and PNR. CH_4_ was significantly and positively correlated with the SOC. CH_4_ was significantly and negatively correlated with the MTN and PM. N_2_O was highly significantly and positively correlated with the MTN. N_2_O was significantly and positively correlated with the PM. N_2_O was significantly and negatively correlated with the PPS, SDR, PCR, and PNR.

## 3. Discussion

### 3.1. Effects of Different Years of Straw Return on the Soil Fertility of Paddy Fields

During the decomposition of rice straw, various nutrients such as carbon, nitrogen, phosphorus, potassium, zinc, iron, and silicon were released into the soil with the involvement of soil microorganisms [29]. The straw decomposition rate was influenced by the soil environment, and the resulting products of decomposition played a role in regulating the soil physiochemical properties. These properties include soil pH, electrical conductivity, as well as the capacity for soil carbon and nitrogen fixation and mineralization [29]. Our results show that the straw decomposition rate, percentage of carbon release, and percentage of nitrogen release increased with increasing years of straw return (within four years of straw return). This was mainly due to the increase in the rate of microorganism multiplication, the number of cellulolytic bacteria, and the soil invertase activity with increasing years of straw return [30].

In the initial stages of straw decomposition, soil microorganisms consume and fix a significant amount of nitrogen, leading to a reduction in the availability of soil nitrogen. This can result in competition between the microorganisms and rice plants for nutrients [31]. Our results indicate that the impact of straw return on soil nutrient availability differs depending on the specific straw return treatments and growth stages of rice, as shown in Figure 8. Compared with the S0 treatment, the soil NO_3_^−^-N and NH_4_^+^-N contents at the TS were lower in the treatment of straw return (S1, S2, S3, and S4) (Figure 8). The NO_3_^−^-N and NH_4_^+^-N contents of the treatment of straw return increased with increasing years of straw return at the middle tillering stage. However, from the jointing stage, the soil NO_3_^−^-N and NH_4_^+^-N contents were higher in the straw return treatments than in the S0 treatment (Figure 8), which would then contribute to higher nitrogen uptake by plants, more biomass, and higher grain yield in the straw return treatments (Figure 1, Table 1). The increase in grain yield was previously observed to be mainly as a result of the reduced risk of nitrogen volatilization during the early vegetative stage and increased spikelet differentiation during the panicle initiation stage [32]. In this sense, the effect of long-term straw return on rice production may involve similar physiological mechanisms. Specifically, long-term straw return can help synchronize crop nitrogen demand with the nitrogen supply from the soil during the rice growing season, particularly when starting from the panicle initiation stage. The practice of straw return has long been advocated and implemented to enhance soil carbon and nitrogen levels [33]. This study found that straw return had a significant positive impact on the soil organic carbon and total nitrogen content. Furthermore, it was observed that the levels of soil organic carbon and total nitrogen increased with increasing years of straw return. These results are consistent with previous studies conducted by Ran et al. (2022) [31]. Under the same year of the straw return condition, the soil organic carbon content and soil total nitrogen content of the treatments in 2021 were higher than those of the treatments in 2022; this might be mainly due to the high rainfall and high temperature in 2021, which accelerated the straw decomposition and nutrient release. Soil organic carbon is the result of microbial activity in the soil and plays a crucial role in regulating both abiotic and biotic properties of the soil. Its content is closely linked to soil quality and agricultural productivity [34]. Studies have shown that there is a strong correlation between increases in soil organic carbon and soil total nitrogen content when straw is returned to the soil [35]. This correlation is attributed to the cycling process of the C/N ratio, which plays a significant role in maintaining a stable state of soil biochemical levels [29,36]. In this study, it was found that straw return improved the soil C/N ratio, mainly due to the fact that the carbon release rate from straw was higher than the nitrogen release rate, and the phenomena of carbon fixation and increased nitrogen mineralization were common in the straw return treatments at high C:N ratios, which might be mainly due to the different soil C/N ratios caused by the different straw return treatments [37].

### 3.2. Effects of Different Years of Straw Return on the Rice Nitrogen Uptake and Grain Yield

Among the yield components that contribute to rice yield, the spikelets per panicle have shown significant potential for improvement. This is because the aforementioned rate exhibits a large degree of variability and adjustability. Increasing the spikelets per panicle has consistently been a primary strategy for achieving higher grain yields in both breeding and cultivation practices [38,39]. We observed that S1 significantly decreased grain yield, which was mainly due to the decrease in panicles per m^2^ when compared with S0 (Table 1). There are several reasons for the decrease in panicles per m^2^. Firstly, the high C/N ratio of rice straw leads to competition for nutrients between microbes and plants, ultimately affecting the growth of rice plants [12]. Secondly, the accumulation of microbial allelochemicals, such as organic acids, from the returned rice straw restricts rice root development [13]. Additionally, our research has shown a significant decrease in enzymes related to the nitrogen metabolism (NR, GOGAT, and GS) and root activity in roots due to straw return (Figure 6). This could be attributed to the lower oxygen content in the soil during the initial 0–40 days after rice transplanting. According to our findings, the delayed revival and slow growth of rice in straw-returning fields can be attributed to the sluggish root growth and decreased activity of enzymes associated with nitrogen metabolism (Figure 7). It is noteworthy that the rice yield showed an upward trend with increasing years of straw return, which aligns with the findings reported by Fan et al. (2022) [40]. When the years of straw return was greater than one year, there was no significant effect of straw return on the rice yield, and this was mainly due to the negative effect of the decrease in the panicles per m^2^ balanced with the positive benefit of the increase in the spikelets per panicle. One of the reasons for the increase in the rice yield was that, with the increase in the years of straw return, the inhibitory effect of the straw return on the pre-growth period of rice was weakened, the negative effect on the tillering of rice was reduced, and the panicles per m^2^ of rice was ensured (Figure 3). Another reason for the increase in yield was that, with increasing years of straw return, the panicle length, the panicle weight, and the spikelets per panicle showed a yearly increase (Table 1, Figure 1). The results of this study demonstrate that the soil inorganic nitrogen content (NO_3_^−^-N and NH_4_^+^-N contents) from the jointing stage of rice was significantly higher in the straw-return treatments compared to S0. This resulted in increased nitrogen uptake by plants and a higher spikelets per panicle (Table 1, Figure 2 and Figure 8). Additionally, the increased nitrogen uptake during the jointing stage in the straw-return treatments led to an increase in the leaf area index and leaf area duration, resulting in higher canopy photosynthesis and photosynthetic assimilation accumulation (Figure 2A,B, Figure 4 and Figure 5). Therefore, it is strongly recommended to provide a higher supply of available nitrogen during the jointing stage to increase the spikelets per panicle in rice production.

### 3.3. Effects of Different Years Straw Return on Greenhouse Gas Emissions

A part of the CO_2_ emissions from paddy fields is generated through plant respiration, while the remaining portion is produced by the metabolic activities of organisms and microorganisms in the soil [41]. The findings of this study indicate that there was a significant peak in the CO_2_ fluxes during the mid-season drainage phase in all treatments. The peak occurred due to the increased soil aeration at the mid-season drainage, which provided oxygen to the soil microorganisms, which are active and release more CO_2_ [35]. Along with reflooding after mid-season drainage, the oxygen in the paddy soil is reduced, the respiration rate of the root system and soil microorganisms slow down, and the CO_2_ emissions decrease [42]. Our results indicate that straw return has a direct impact on the increase in CO_2_ emission from paddy fields, which aligns with the results obtained by Han et al. (2023) [41]. Additionally, Tang et al. (2022) found a close relationship between soil organic carbon and CO_2_ emission and absorption in paddy fields [43]. The process of straw return leads to an increase in soil organic carbon and soil C/N, which, in turn, enhances soil microbial respiration. Furthermore, Chen et al. (2021) discovered that straw, acting as an exogenous carbon source, stimulates the release of CO_2_ by triggering the decomposition of natural organic carbon [32]. In addition, correlation analysis showed that the CO_2_ accumulation was negatively correlated with the maximum number of tillers and the panicles per m^2^ of rice (Figure 13). Compared with S0, the straw return reduced the maximum number of tillers and panicles per m^2^ of rice (Figure 13). Therefore, the straw-return treatment increased the CO_2_ emission of the rice. The present study found no significant difference in the emission flux of the CO_2_ from paddy fields between different years of straw return, which is consistent with the findings of Zhang et al. (2015) [44].

CH_4_ emissions in rice paddies are primarily influenced by methanogens, which produce CH_4_ under anaerobic conditions. Methanogens are affected by the availability of soil carbon [22]. Straw contains cellulose and hemicellulose, which serve as carbon substrates for methanogenic bacteria, leading to increased CH_4_ production [45]. Consistent with previous studies, straw return has been found to significantly increase CH_4_ emissions [33]. Additionally, the presence of higher soil organic carbon in straw-return treatments suggests greater decomposition (Table 2, Figure 9 and Figure 10). The higher CH_4_ emissions under straw return may be attributed to the inhibition of methane-oxidizing bacteria activity, which occurs due to microbial O_2_ consumption [33]. In this study, we found no significant difference in the emission fluxes of the CH_4_ from paddy fields that were subjected to different years of straw return, which is inconsistent with the findings of Huang et al. (2022). This was mainly due to the differences in the years of straw return [20]. In addition, correlation analysis showed that the CH_4_ accumulation was highly significantly and negatively correlated with the maximum number of tillers and the panicles per m^2^ of rice, suggesting that increasing the panicles per m^2^ of rice can reduce greenhouse gas emissions (Figure 13).

N_2_O is the intermediate product of soil bacteria’s nitrification and denitrification processes [24]. In this study, it was found that straw return did not have an impact on N_2_O emissions [46]. We propose three possible explanations for these findings. Firstly, when the C/N ratio of the straw was between 20–75, the fixation of mineral nitrogen and the release of soil available nitrogen from straw decomposition reached a dynamic equilibrium. In our study, the C/N ratio of the rice straw returned to the fields was 60.70 [47], which means that very little nitrogen was converted into N_2_O. Secondly, straw return undoubtedly and easily provides labile carbon and nitrogen substrates that promote the growth of nitrification and denitrification-related microorganisms, and it subsequently enhances the potential for nitrification and denitrification [22]. The improvement of nitrification due to increased reaction substrates from the decomposition of returned straws might promote N_2_O emissions. However, it is important to note that, while straw return can increase the concentration of readily labile organic carbon, a high concentration of labile organic carbon can actually decrease N_2_O/N_2_ ratios, leading to a decrease in N_2_O emissions [48]. Additionally, the decomposition of straws can consume soil oxygen and create anaerobic conditions in the soil [47]. This can enhance denitrification, resulting in a large quantity of N_2_O being converted to N_2_. Therefore, it is possible that no significant effects of straw return on N_2_O emissions might be observed in this study.

In this study, the GWP of the straw-return treatments were significantly higher in S0, and there was no significant difference found between the different straw-return treatments. The order of the GHGI regarding significant impact on the GWP was CH_4_ > CO_2_ > N_2_O. Therefore, the differences in the GWP between treatments were more closely aligned with the variations in CH4 emissions. Compared with S0, the percentage increase in the GHGI of S1, S2, and S3 was greater than the percentage increase in the GWP. This was because the reduction in rice yield due to varying degrees of straw-return conditions led to an increase in the GHGI. Compared with S0, the percentage increase in the GHGI of S4 was smaller than the percentage increase in the GWP. This was because the increase in the rice yield under S4 led to a decrease in the GHGI (Figure 14).

## 4. Materials and Methods

### 4.1. Experimental Site and Soil Properties

An experiment was conducted at the National Crop Variety Validation Characterization Station on the campus of Jilin Agricultural University in Changchun, Jilin Province, China (43°48′ N, 125°24′ E), from early May to early October in both 2021 and 2022. The region has a continental monsoon climate. The total accumulated temperature and rainfall during the reproductive period were 2991 °C and 860.3 mm in 2021 and 2800 °C and 600 mm in 2022, respectively. The daily mean air temperature and precipitation during the rice-growing seasons are shown in Figure 15. The trial plot was composed of phaeozem soil and had an organic carbon content of the soil of 9.60 g kg^−1^. The alkaline dissolved nitrogen was 33.89 mg kg^−1^; the available phosphorus was 29.42 mg kg^−1^; the available potassium was 137.09 mg kg^−1^; and the pH was 6.7.

### 4.2. Experimental Design

This experiment was conducted from 10 May 2019 to 5 October 2022. Employing a randomized complete block design, this study comprised three replications. The experiment was designed with four treatments in 2021 with no straw return (S0), one year of straw return (S1), two successive years of straw return (S2), and three successive years of straw return (S3). In 2022, there will be one additional treatment compared to 2021: four successive years of straw return (S4). The size of each treatment plot was 5 m × 6 m = 30 m^2^.

The amount of straw return was 4500 kg hm^−2^. Rice straw was generated from the previous rice cultivation season. After air drying under natural conditions, the straw was chopped into 5–7 cm pieces. Rototilling was conducted annually in mid-May. In subplots where the straw was returned, it was evenly spread on the soil surface and incorporated into the soil using a reverse stubble rotavator. During the rice-growth season, all treatments applied 175 kg hm^−2^ of pure nitrogen (urea) in a ratio of 6:3:1 (basal fertilizer:tiller fertilizer:panicle fertilizer). Furthermore, all treatments, respectively, applied 75 kg hm^−2^ of P_2_O_5_ (calcium superphosphate) and K_2_O (potassium chloride) as the base fertilizer.

The experiment utilized the Jinongda 667 variety of japonica rice. Tillage was conducted annually on 10 May, followed by ponding on 12 May and the transplantation of seedlings on 20 May. The spacing between rows was 30 cm, with 5 seedlings per hill placed 13.3 cm apart. On 16 May 2022, nylon mesh bags filled with rice straw were buried at the field’s center, with a marker set at the site [47]. The mesh bags were positioned vertically without overlapping horizontally, with a distance of 5–10 cm from the soil layer. The mesh bags, each 25 cm long and 20 cm wide, had a standard mesh size with a 0.154 mm diameter. The straw samples, cut to about 5 cm length and air-dried, were placed in the mesh bags and sealed using nylon thread. Each bag contained 30.0 g of air-dried straw, equivalent to 28.17 g in oven-dry mass. Harvesting was performed around October 1st in both years. The trial plots were managed separately for irrigation and drainage, following local high-yielding cultivation practices. Cement ridges were utilized to prevent seepage between plots and exchange in moisture and fertilizers.

### 4.3. Measurement Index and Methods

#### 4.3.1. Rice Yield

At maturity, we selected 5 representative points of 1 m^2^ and calculated the panicles per m^2^. And we selected 10 hills with the same number of the panicles from 5 points for drying, then calculated the spikelets per panicle, 1000-grain weight, and filled grain rate. In addition, five random sampling points were chosen within each plot (excluding edge rows and initial sampling rows), and a total area of 3 m^2^ was harvested at each point to assess the yield per unit area. Each plot was harvested individually, and the moisture content of the rice grains was measured following the drying process. The final grain yield was calculated by adjusting the obtained data to a standard moisture content of 13.5%.

#### 4.3.2. Leaf Area Index, Dry Weight, and Nitrogen Uptake of Rice

Five rice plants were sampled from each plot at the middle tillering stage (TS), jointing stage (JS), heading stage (HS), filling stage (FS), and maturity stage (MS). The samples of the above-ground parts of the plants were divided into leaf, stem, and spikelet. The oven-drying method was utilized at 80 °C to ascertain the dry weight of each component. An area meter (LAI-2200C, Beijing Ecotek Technology Company Limited, Beijing, China) was employed to measure the leaf area index. Leaf area duration (LAD) was calculated by the following equation:(1)$$\mathrm{LAD}\;(m^{2}\;m^{-2}\;d)=\frac{1}{2}\times (L1+L2)\times (t2-t1),$$
where L1 and L2 are the leaf area index (LAl, m^2^ m^−2^) measured at time t1 and t2, respectively.

The nitrogen uptake per plant was calculated by multiplying the nitrogen concentration per plant (%) by the biomass per plant, and the nitrogen concentration was determined using the micro Kjeldahl method [49]. The plant nitrogen uptake of the above ground biomass was calculated (concentration of nitrogen in plants × above ground biomass dry weight). One replicate for the measurement consisted of sampling five plants per plot, with a total of five replicates for each treatment.

The panicle length and panicle weight were measured using a ruler and an electronic balance, respectively. The number of tiller growths was monitored in each plot every 7 days until the rice plants reached the heading stage.

#### 4.3.3. Measurement of the Photosynthetic Parameters and SPAD Value of the Rice Leaves

At the TS, JS, HS, FS, and MS, the SPAD values and net photosynthetic rate (Pn) of rice flag leaves were measured. Five points were selected in each plot (excluding the side rows), and, at each point, five rice plants with the same growth trend were chosen based on the average number of tillers. The Pn of the flag leaves was measured using a Li-6400 photosynthesizer (Li-Cor Inc., Lincoln, Dearborn, MI, USA) manufactured in the USA between 9:00–11:00 a.m. on a sunny day without wind. The SPAD values, which indirectly indicate leaf chlorophyll content [50], were measured using a SPAD-502 portable chlorophyll meter (Minolta, Osaka, Japan) for the chlorophyll content of the flag leaves. Each measurement was repeated five times.

#### 4.3.4. Rice Root Sample Collection and Measurement

At 10, 20, 30, and 40 days after transplanting, ten hill samples were collected from each plot based on the average tiller count. Five samples were utilized for analyzing root morphological traits, while the remaining five were dedicated to studying root physiological traits. For each root sampling, a cube of soil (30 cm long × 16.5 cm wide × 40 cm deep) was removed around each individual hill with a sampling core. This cube encompassed approximately 95% of the total root biomass [51]. The roots within each soil block were meticulously washed using a hydro-pneumatic elutriation apparatus (Gillison’s Variety Fabrications. Benzonia, MI, USA). Root length was measured by a standard ruler. The number of roots was counted manually. The BC0080 and BC0070 test kits from Beijing Solaibao Technology Co., Ltd. (Beijing, China) were used to determine the amount of nitrate reductase (NR) and glutamate synthase (GOGAT) by UV spectrophotometry, and the BC0910 test kit was used to determine the glutamine synthase (GS) activity by visible spectrophotometry.

#### 4.3.5. Measurement of Straw Decomposition

At the TS, JS, HS, FS, and MS, sampling of the nylon bags was performed. Thus, a total of 5 samplings were conducted, with 5 bags per sampling. Following the collection of the samples, they were rinsed and cleaned to remove any rice roots that had infiltrated the net bags. The samples were subsequently dried in an oven, weighed, crushed, and then transferred to plastic sealing bag for future use.

The carbon content was measured both by oxidation with potassium dichromate [49]. The total nitrogen content was measured by digestion with H_2_SO_4_–H_2_O_2_ following the Kjeldahl method (The Foss Group Corporation of Denmark, Hilloerod, Denmark) [49].

The following calculations was performed:(2)$$\mathrm{Straw}\;\mathrm{decomposition}\;\mathrm{rate}=\frac{\left(m0-m1\right)}{m0}\times \;100\%,$$
where m0 is the initial straw mass or nutrient content, and m1 is the mass or nutrient content in the straw residue on a given sampling date.

#### 4.3.6. Soil Sample Collection and Determination

At the TS, JS, HS, FS, and MS, soil samples were collected from all plots at a depth of 0–20 cm utilizing a 2.5 cm corer. Five points within each plot were sampled, with a distance of 2.5 m between each sampling point. Subsequently, these samples were combined to produce a composite sample weighing about 500 g. The composite samples were stored in a plastic sealing bag and transported to the laboratory for further processing. Prior to an analysis of soil chemical traits, the plant remains, roots, and rocks were manually removed. The samples were air-dried and pulverized before analysis.

K_2_Cr_2_O_7_-H_2_SO_4_ oxidation was used to measure soil organic carbon [49]. The total nitrogen content was determined using the micro-Kjeldahl method [49]. The lixiviation of NO_3_^−^-N and NH_4_^+^-N from 1 M of KCl was used as a flow-injection analyzer (SAN++; Skalar Analytical B.V., Breda, The Netherlands). The Olsen method with 0.5 M of NaHCO_3_ extraction was used to measure the available phosphorus [49]. The flame photometric method was utilized to determine the available potassium [49].

#### 4.3.7. Measurement of Soil GHG Emissions

The measurement of CO_2_, CH_4_, and N_2_O fluxes was conducted through closed box-gas chromatography. Samples of gas were gathered weekly following the transplanting of rice seedlings, with the sampling taking place at 9–11 a.m. If there was substantial rainfall, the sampling time was delayed. The chamber utilized in this study had measurements of 20 cm × 40 cm × 120 cm (width × long × height) and was furnished with a ventilator for gas blending and a thermostat for observing air temperature. Each plot involved securing a chamber base (20 cm width × 40 cm long × 20 cm height) in the soil at a depth of 15 cm, covering two rice plant hills. When collecting samples, the gap between the groove and the chamber was water-sealed. At each plot, samples (30 mL) were injected into pre-evacuated vials with a syringe at 0, 15, 30, and 45 min. The GHG (CO_2_, N_2_O, CH_4_) concentrations were determined by a gas chromatograph (Agilent 7890A, Agilent Technologies, Santa Clara, CA, USA). The emission rates of the greenhouse gas (F) were calculated as follows [21]:(3)$$F=\frac{\Delta C}{\Delta T}\times \frac{V}{A},$$
where ΔC/ΔT is the change in greenhouse gas (CO_2_, N_2_O, and CH_4_) concentration (mg L^−1^ h^−1^) in the chamber, as determined by linear regression; V is the volume of the chamber (L); and A is the enclosed surface area (m^2^). For our flux rate estimates, we only accepted measurements for which r^2^ > 0.90; as such, less than 5% of the measurements were discarded.

The cumulative GHG emissions were based on two adjacent sampling periods, while gas emissions for the two adjacent sampling periods were understood as the product of the average emission flux and the sampling time [52].

For assessing the effects of GHG mitigation, a total gross GWP in CO_2_-e per hectare was calculated using the following equation [53]:GWP = E(CO_2_) + E(CH_4_) × 25 + E(N_2_O) × 298, (4)
where E(CO_2_), E(CH_4_), and E(N_2_O) are the seasonal totals of CO_2_, CH_4_, and N_2_O emission (kg hm^−2^) monitored in a single cycle, respectively. The default molecular GWP of the CH_4_ and N_2_O in a 100-year time frame as 25 and 298 was used in the calculation, respectively, while the GWP value for CO_2_ was taken as 1 [54].

The GHG emission intensity was calculated as follows:(5)$$\mathrm{GHGI}=\frac{\mathrm{GWP}}{\mathrm{RY}},$$
where GHGI is the GHG emission intensity (kg CO_2_-e kg^−1^ yield), and RY is the rice yield (kg hm^−2^).

### 4.4. Statistical Analysis

IBM SPSS Statistics 26 software was used for statistical analysis, in which the analysis of variance (ANOVA) was one-way ANOVA, Duncan’s method was used for multiple comparisons, and the differences between the means were statistically significant at the level of <0.05. The mean values with standard errors (SE) were reported for all data. Figures were created using Origin 2023 software. Data organization and table generation were performed using Microsoft Excel 2021.

## 5. Conclusions

Successive years of straw return were found to have several positive effects on paddy fields. Firstly, successive years of straw return increased the root length, the number of roots, and the activity of nitrogen metabolizing enzymes in the root system at the tillering stage of rice, which, in turn, promoted the occurrence of rice tillers and increased the panicles per m^2^. Additionally, successive years of straw return helps to promote straw decomposition; increase soil nutrients (soil organic carbon, total nitrogen, NH_4_^+^-N, NO_3_^−^-N, available phosphorus, and available potassium) in paddy fields; synchronize the supply of nitrogen in the soil with the demand of rice plants; improve photosynthetic capacity from the jointing stage to maturity stage; and increase the panicle length, panicle weight, and spikelets per panicle. These combined effects ultimately promote the formation of rice yields. Moreover, successive years of straw return will decrease the GHGI (Figure 14), indicating its potential in mitigating greenhouse gas emissions. The results of this study can provide a valuable theoretical basis and technical support for the rational use of straw, the protection of black soil, the improvement of crop yields, and the achievement of sustainable agricultural development in the northeast region of China.

## Figures and Tables

**Figure 1 plants-13-02446-f001:**
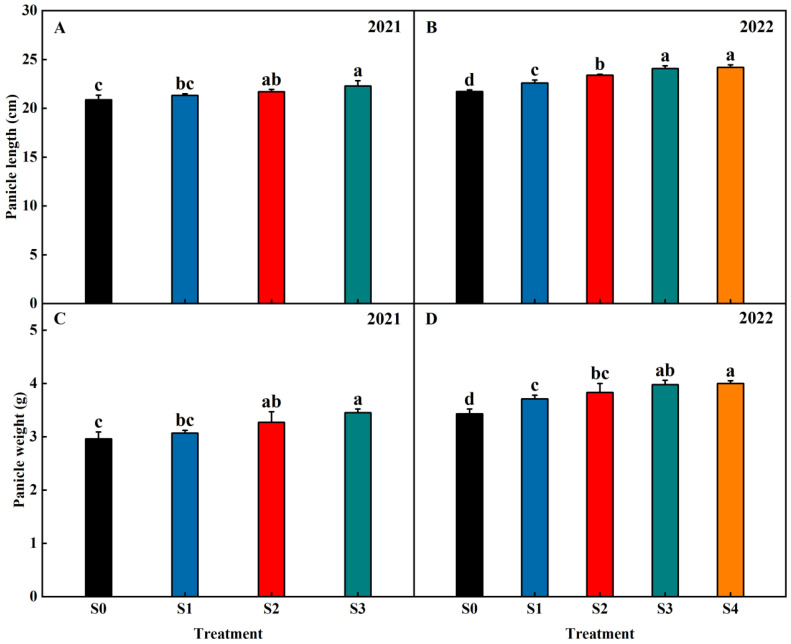
Effects of different years of straw return on the panicle length (**A**,**B**) and panicle weight (**C**,**D**) of rice. Different letters labeling the bars indicate statistical significance at the *p* < 0.05 level. S0, S1, S2, S3, and S4 are no straw return, one year of straw return, two successive years of straw return, three successive years of straw return, and four successive years of straw return, respectively.

**Figure 2 plants-13-02446-f002:**
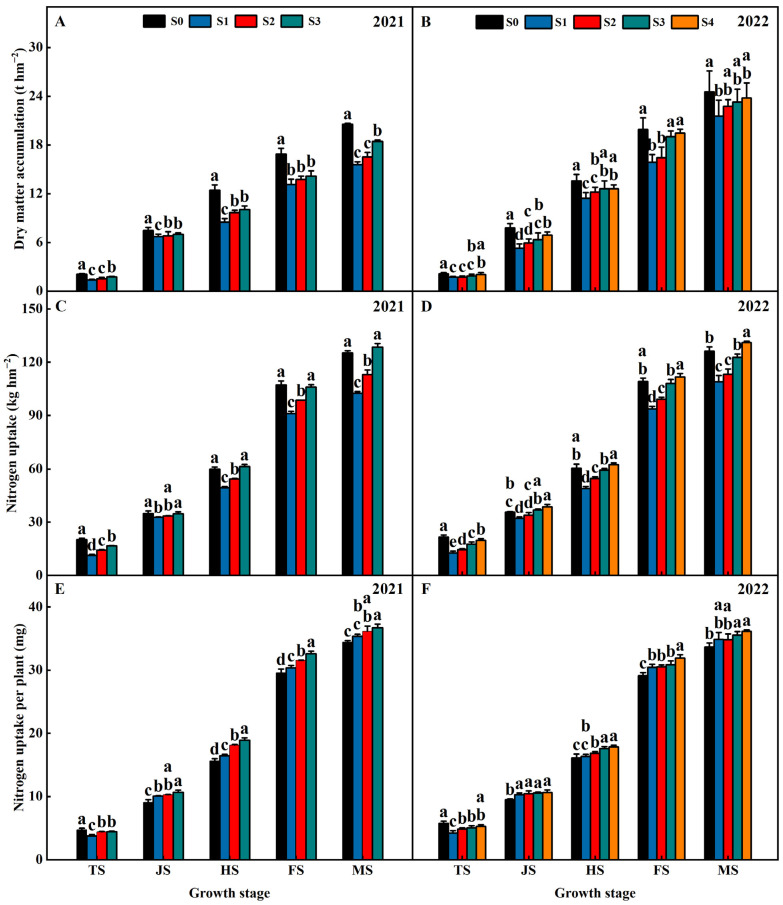
Effects of different years of straw return on the dry matter accumulation (**A**,**B**), nitrogen uptake (**C**,**D**), and nitrogen uptake per plant (**E**,**F**) of rice. Different letters labeling the bars within the same growth period indicate statistical significance at the *p* < 0.05 level. S0, S1, S2, S3, and S4 are no straw return, one year of straw return, two successive years of straw return, three successive years of straw return, and four successive years of straw return, respectively. TS, JS, HS, FS, and MS are the middle tillering stage, jointing stage, heading stage, filling stage, and maturity stage, respectively.

**Figure 3 plants-13-02446-f003:**
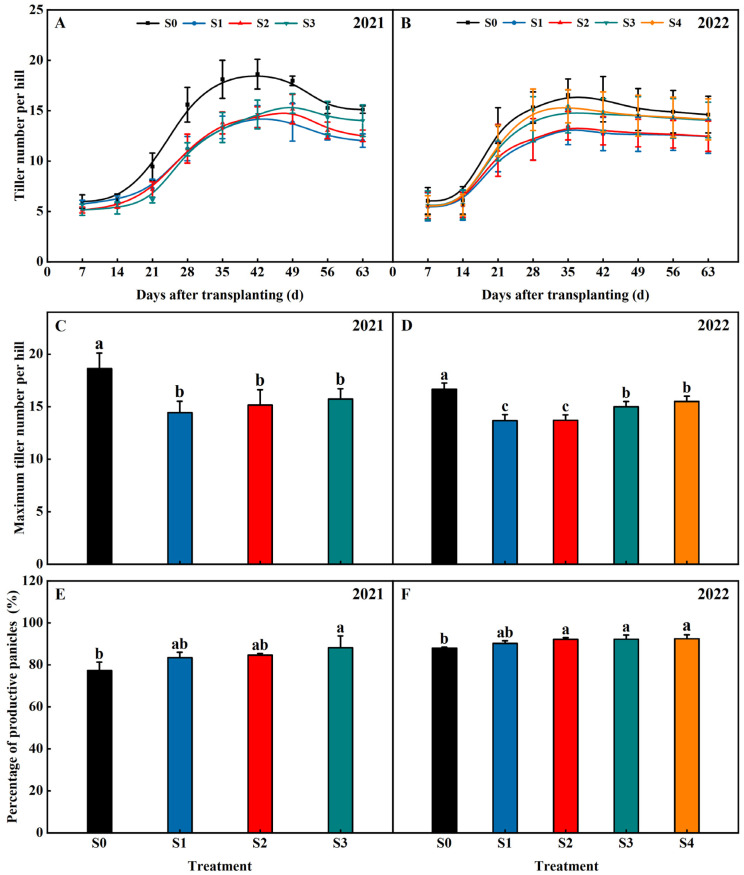
Effects of different years of straw return on tiller number per hill (**A**,**B**), maximum tiller number per hill (**C**,**D**), and percentage of productive panicles (**E**,**F**) of rice. Different letters labeling the bars indicate statistical significance at the *p* < 0.05 level. S0, S1, S2, S3, and S4 are no straw return, one year of straw return, two successive years of straw return, three successive years of straw return, and four successive years of straw return, respectively.

**Figure 4 plants-13-02446-f004:**
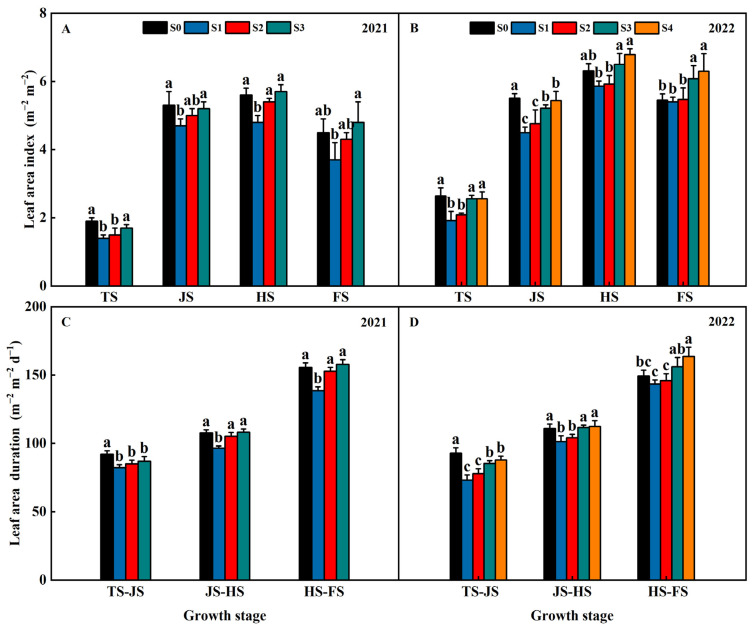
Effects of different years of straw incorporation on leaf area index (**A**,**B**) and leaf area duration (**C**,**D**) of rice flag leaf. Different letters labeling the bars within the same growth period indicate statistical significance at the *p* < 0.05 level. S0, S1, S2, S3, and S4 are no straw return, one year of straw return, two successive years of straw return, three successive years of straw return, and four successive years of straw return, respectively. TS, JS, HS, FS, and MS are the middle tillering stage, jointing stage, heading stage, filling stage, and maturity stage, respectively.

**Figure 5 plants-13-02446-f005:**
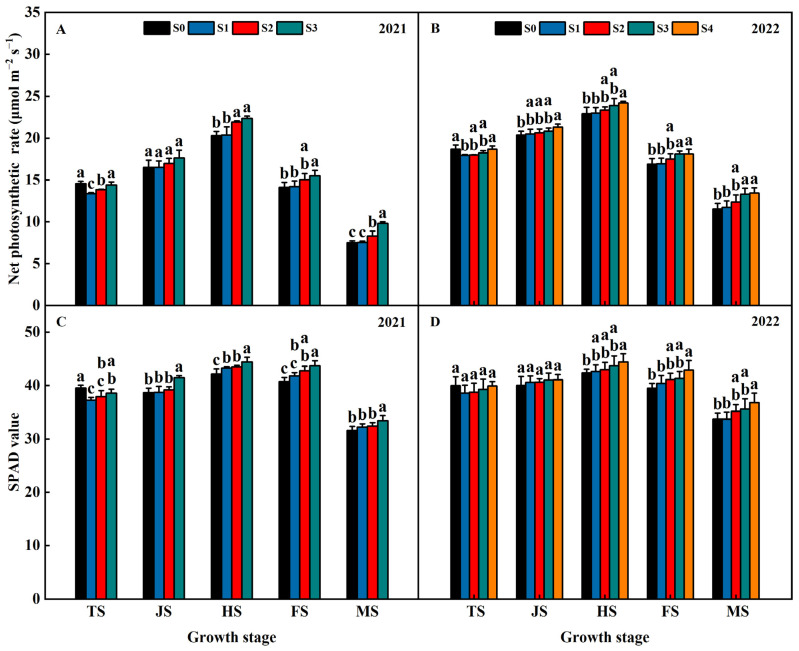
Effects of different years of straw return on Pn (**A**,**B**) and SPAD value (**C**,**D**) of rice flag leaf. Different letters labeling the bars within the same growth period indicate statistical significance at the *p* < 0.05 level. S0, S1, S2, S3, and S4 are no straw return, one year of straw return, two successive years of straw return, three successive years of straw return, and four successive years of straw return, respectively. TS, JS, HS, FS, and MS are the middle tillering stage, jointing stage, heading stage, filling stage, and maturity stage, respectively.

**Figure 6 plants-13-02446-f006:**
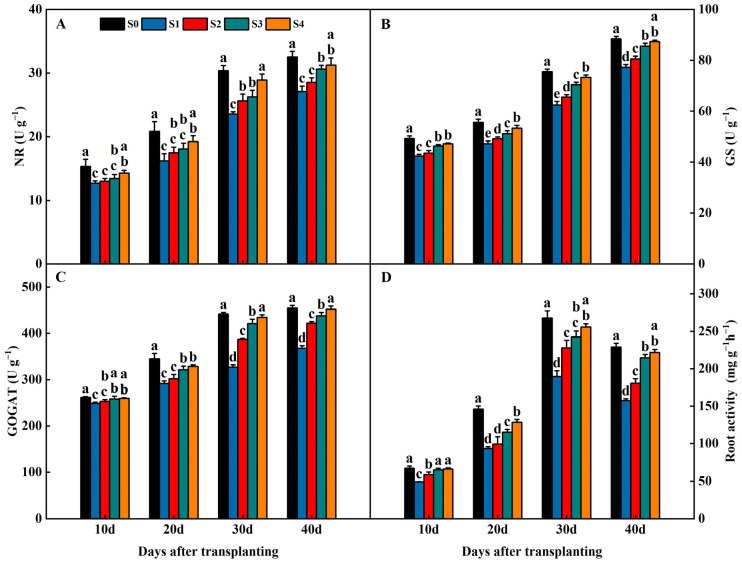
Effects of different years of straw return on the activity of nitrogen invertase and root activity in 2022. (**A**–**D**) are NR (nitrate reductase), GS (glutamine synthase), GOGAT (glutamate synthase), and root activity, respectively. Different letters labeling the bars within the same growth period indicate statistical significance at the *p* < 0.05 level. S0, S1, S2, S3, and S4 are no straw return, one year of straw return, two successive years of straw return, three successive years of straw return, and four successive years of straw return, respectively.

**Figure 7 plants-13-02446-f007:**
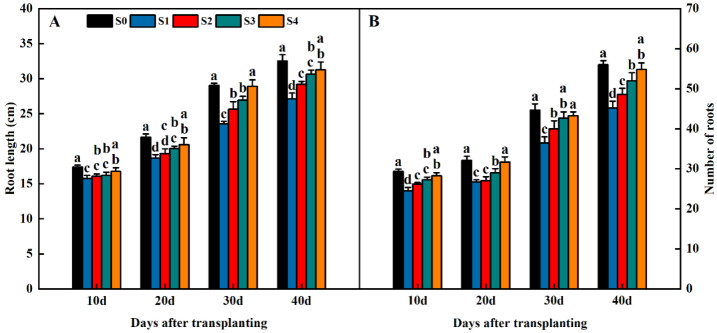
Effects of different years of straw return on the root length (**A**) and number of roots (**B**) in 2022. Different letters labeling the bars within the same growth period indicate statistical significance at the *p* < 0.05 level. S0, S1, S2, S3, and S4 are no straw return, one year of straw return, two successive years of straw return, three successive years of straw return, and four successive years of straw return, respectively.

**Figure 8 plants-13-02446-f008:**
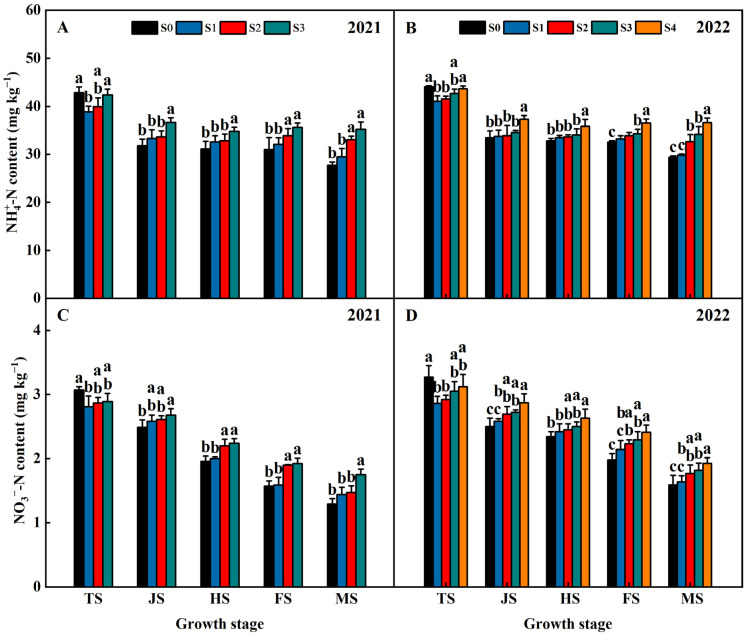
Effects of different years of straw return on the NH_4_^+^-N (**A**,**B**) and NO_3_^−^-N (**C**,**D**) contents. Different letters labeling bars within the same growth period indicate statistical significance at the *p* < 0.05 level. S0, S1, S2, S3, and S4 are no straw return, one year of straw return, two successive years of straw return, three successive years of straw return, and four successive years of straw return, respectively. TS, JS, HS, FS, and MS are the middle tillering stage, jointing stage, heading stage, filling stage, and maturity stage, respectively.

**Figure 9 plants-13-02446-f009:**
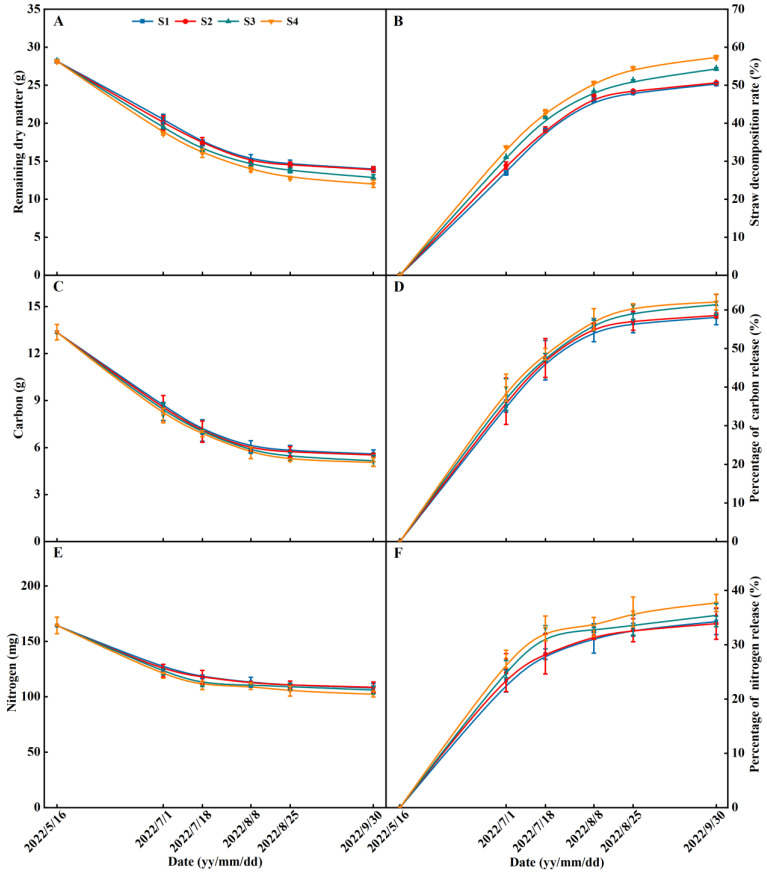
Effects of different years of straw return on the remaining dry matter, carbon, and nitrogen, and their percentage of nutrient release of straw in 2022. (**A**–**F**) are the remaining dry matter, straw decomposition rate, carbon, percentage of carbon release, nitrogen, and percentage of nitrogen release, respectively. S0, S1, S2, S3, and S4 are no straw return, one year of straw return, two successive years of straw return, three successive years of straw return, and four successive years of straw return, respectively. The TS (1 July 2022), JS (18 July 2022), HS (8 August 2022), FS (25 August 2022), and MS (30 September 2022) are the middle tillering stage, jointing stage, heading stage, filling stage, and maturity stage, respectively.

**Figure 10 plants-13-02446-f010:**
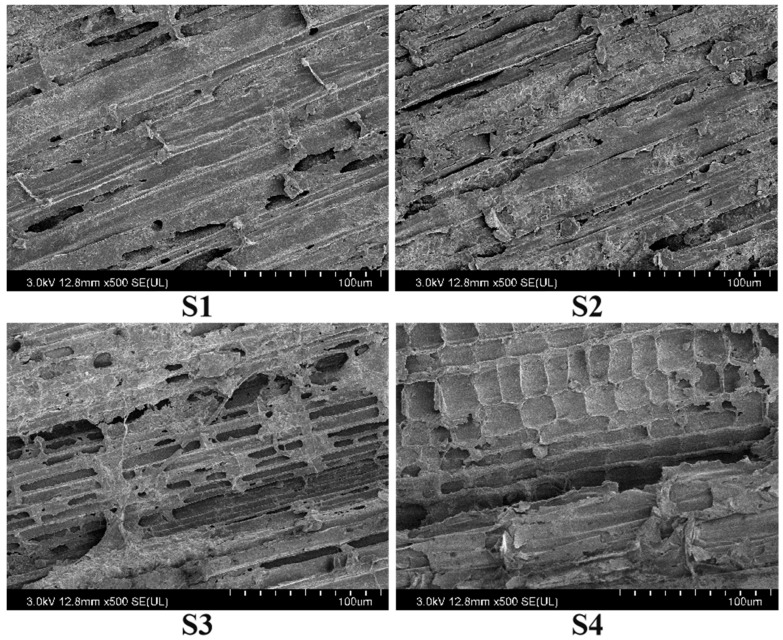
Scanning electron microscope (SEM) images of the decomposition of the straw tillering stage over different years of straw return in 2022. The scale bar is 100 μm per grid. S0, S1, S2, S3, and S4 are one year of straw return, two successive years of straw return, three successive years of straw return, and four successive years of straw return, respectively.

**Figure 11 plants-13-02446-f011:**
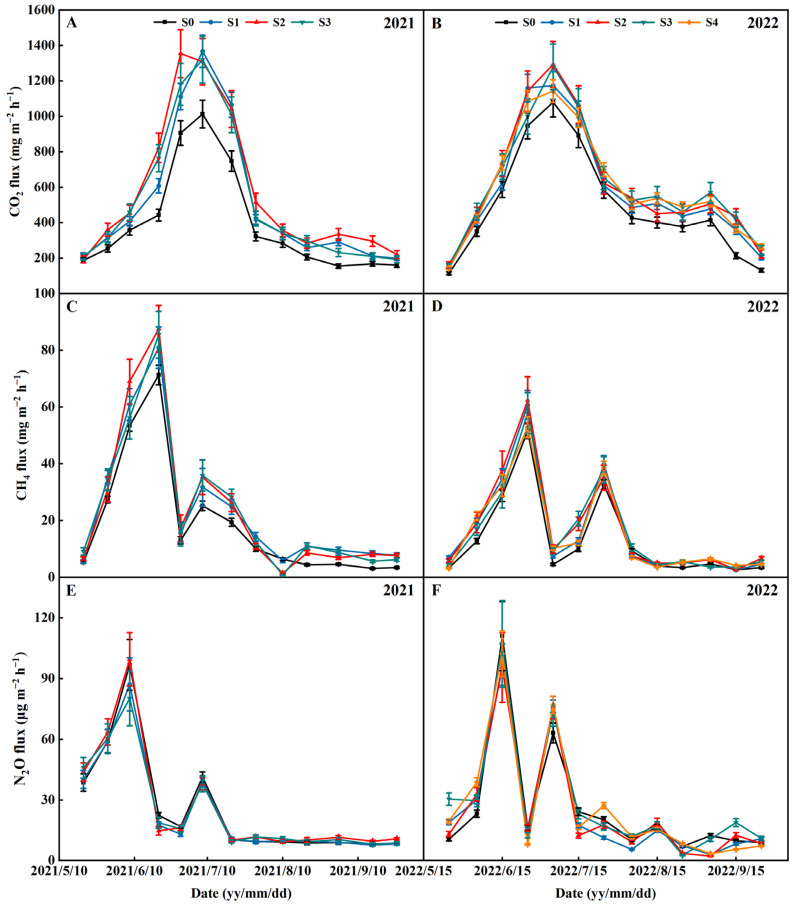
Effects of different years of straw return on the CO_2_ (**A**,**B**), CH_4_ (**C**,**D**), and N_2_O (**E**,**F**) flux. S0, S1, S2, S3, and S4 are no straw return, one year of straw return, two successive years of straw return, three successive years of straw return, and four successive years of straw return, respectively.

**Figure 12 plants-13-02446-f012:**
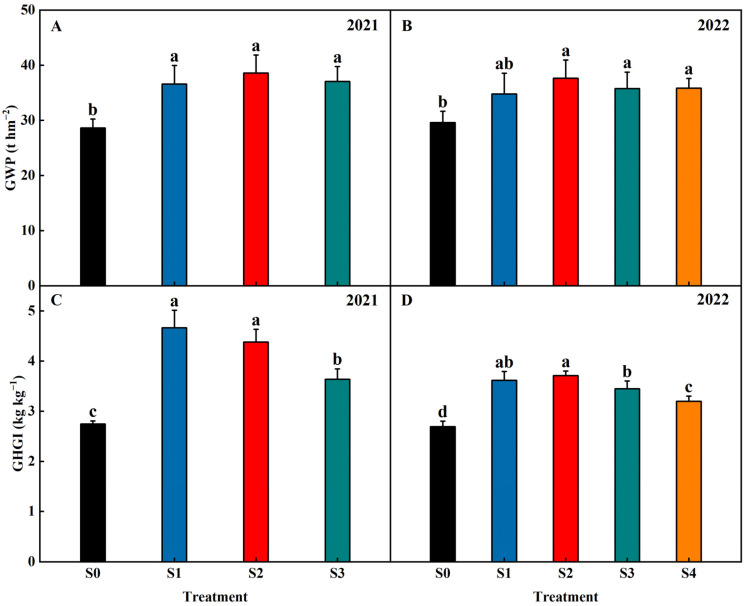
Effects of different years of straw return on GWP (**A**,**B**) and GHGI (**C**,**D**). Different letters labeling the bars indicate statistical significance at the *p* < 0.05 level. S0, S1, S2, S3, and S4 are no straw return, one year of straw return, two successive years of straw return, three successive years of straw return, and four successive years of straw return, respectively.

**Figure 13 plants-13-02446-f013:**
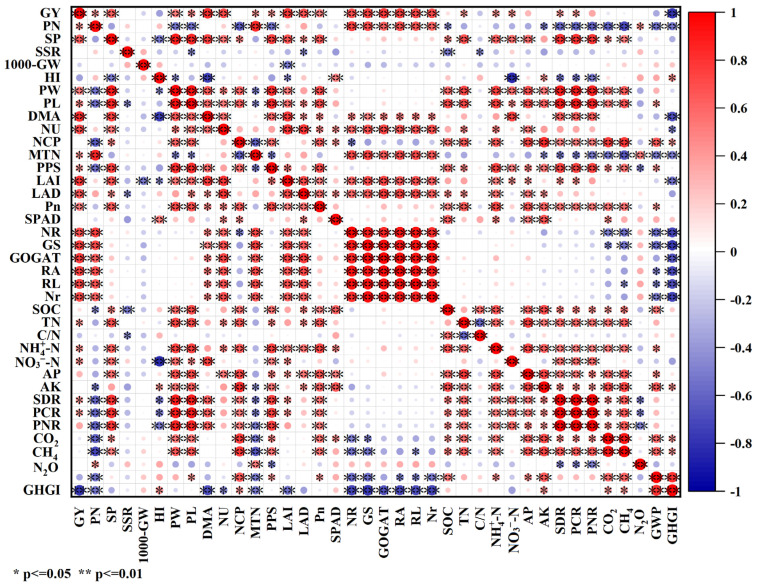
Correlations between the rice yield, CO_2_, CH_4_, N_2_O emissions, and other indexes. RY = rice yield, PM = panicle per m^2^, SP = spikelets per panicle, SSR = seed setting rate, 1000-GW = 1000-grain weight, HI = harvest index, PW = panicle weight, PL = panicle length, DMA = dry matter accumulation, NU = nitrogen uptake, NCP = nitrogen uptake per plant, MTN = maximum tiller number, PPS = percentage of productive spike, LAI = leaf area index, LAD = leaf area duration, Pn = net photosynthetic rate, SPAD = SPAD value, NR = nitrate reductase, GS = glutamine synthase, GOGAT = glutamate synthase, RA = root activity, RL = root length, Nr = number of roots, SOC = soil organic carbon, TN = total nitrogen, C/N = soil C/N ratio, AP = available phosphorus, AK = available potassium, SDR = straw decomposition rate, PCR = percentage of carbon release, PNR = percentage of nitrogen release, CO_2_ = CO_2_ emission, CH_4_ = CH_4_ emission, N_2_O = N_2_O emissions, GWP = global warming potential, and GHGI = greenhouse gas intensity.

**Figure 14 plants-13-02446-f014:**
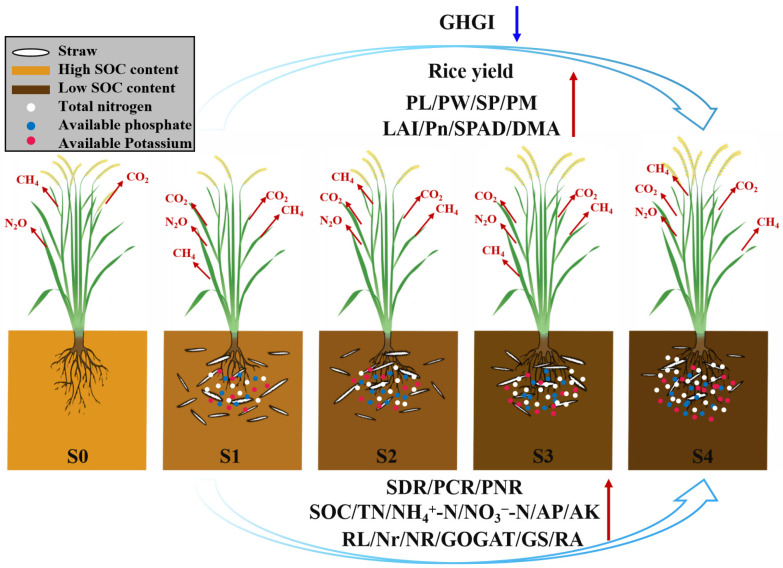
Effects of years of straw return on the rice yield formation with soil nutrients and greenhouse gas emissions. The blue and red arrows represent the decrease and increase, respectively. S0, S1, S2, S3, and S4 are no straw return, one year of straw return, two successive years of straw return, three successive years of straw return, and four successive years of straw return, respectively. GHGI = greenhouse gas intensity, PL = panicle length, PW = panicle weight, SP = spikelets per panicle, PM = panicle per m^2^, LAI = leaf area index, Pn = Net photosynthetic rate, SPAD = SPAD value, DMA = dry matter accumulation, SDR = straw decomposition rate, PCR = percentage of carbon release, PNR = percentage of nitrogen release, SOC = soil organic carbon, TN = total nitrogen, AP = available phosphorus, AK = available potassium, RL = root length, Nr = number of roots, NR = nitrate reductase, GOGAT = glutamate synthase, GS = glutamine synthase, and RA = root activity.

**Figure 15 plants-13-02446-f015:**
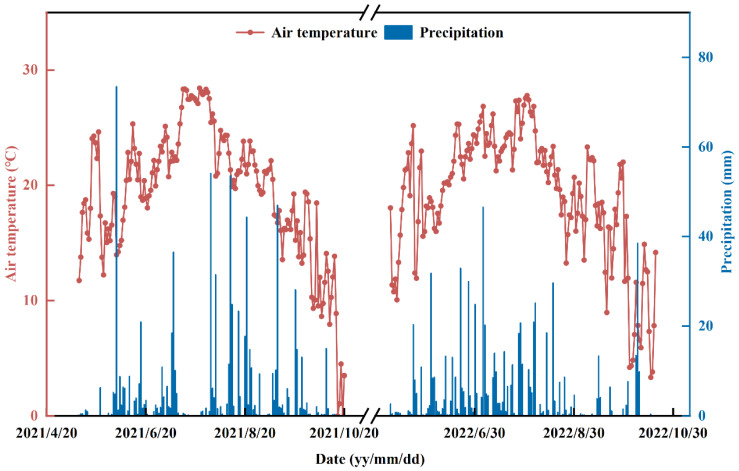
The daily mean air temperature and precipitation during the rice-growing seasons from 2021 to 2022.

**Table 1 plants-13-02446-t001:** Effects of different years of straw return on the rice yield and yield components.

Year	Treatment	Panicles per m^2^	Spikelets per Panicle	Seed Setting Rate (%)	1000-Grain Weight (g)	Yield (t hm^−2^)	Harvest Index (%)
2021	S0	348.33 ± 13.74 a	139.06 ± 5.73 b	94.20 ± 1.46 a	22.33 ± 0.40 a	10.19 ± 0.44 a	50.31 ± 1.06 c
	S1	279.40 ± 10.30 c	141.00 ± 6.49 b	94.66 ± 1.00 a	22.47 ± 0.51 a	8.36 ± 0.28 b	50.79 ± 1.35 c
	S2	307.00 ± 5.24 b	151.87 ± 5.83 a	93.76 ± 1.58 a	22.23 ± 0.32 a	9.72 ± 0.44 a	53.71 ± 1.33 b
	S3	310.80 ± 7.05 b	157.60 ± 4.93 a	93.60 ± 1.63 a	22.06 ± 0.15 a	10.11 ± 0.23 a	55.21 ± 1.07 a
2022	S0	318.33 ± 17.95 a	166.28 ± 3.85 c	95.36 ± 0.54 a	21.27 ± 0.59 a	10.73 ± 0.23 ab	41.67 ± 1.45 b
	S1	267.67 ± 6.03 b	171.22 ± 9.23 bc	95.42 ± 1.82 a	21.96 ± 1.00 a	9.60 ± 0.58 c	42.13 ± 1.63 b
	S2	270.33 ± 16.17 b	178.33 ± 10.86 abc	92.79 ± 5.31 a	22.69 ± 0.97 a	10.13 ± 0.64 bc	45.36 ± 1.36 ab
	S3	273.67 ± 6.03 b	188.11 ± 15.50 ab	92.47 ± 3.46 a	22.23 ± 0.95 a	10.56 ± 0.29 ab	47.21 ± 1.24 a
	S4	292.67 ± 29.70 ab	193.11 ± 10.05 a	91.47 ± 5.13 a	22.54 ± 0.16 a	11.23 ± 0.19 a	47.27 ± 2.53 a

Data are the mean ± SD. Different letters indicate statistical significance at the *p* < 0.05 level within the same column. S0, S1, S2, S3, and S4 are no straw return, one year of straw return, two successive years of straw return, three successive years of straw return, and four successive years of straw return, respectively.

**Table 2 plants-13-02446-t002:** Effects of different years of straw return on the soil chemical traits of paddy rice field.

Year	Treatment	Soil Organic Carbon (g kg^−1^)	Total Nitrogen (g kg^−1^)	C/N Ratio	Available Phosphorus (mg kg^−1^)	Available Potassium (mg kg^−1^)
2021	S0	9.60 ± 0.33 c	1.01 ± 0.02 b	9.26 ± 0.60 a	29.75 ± 0.01 c	138.43 ± 0.00 b
	S1	9.82 ± 0.25 bc	1.04 ± 0.04 ab	9.29 ± 0.38 a	31.08 ± 1.02 bc	149.09 ± 2.31 a
	S2	10.25 ± 0.25 ab	1.07 ± 0.04 ab	9.31 ± 0.72 a	32.41 ± 1.11 ab	151.75 ± 2.31 a
	S3	10.54 ± 0.43 a	1.11 ± 0.05 a	9.41 ± 0.83 a	33.08 ± 1.15 a	154.42 ± 4.00 a
2022	S0	9.71 ± 0.64 b	1.05 ± 0.04 b	9.42 ± 0.89 a	30.75 ± 0.53 b	139.76 ± 2.31 d
	S1	9.98 ± 0.28 ab	1.08 ± 0.03 ab	9.54 ± 0.57 a	31.21 ± 0.70 b	145.09 ± 2.31 cd
	S2	10.3 ± 0.44 ab	1.11 ± 0.04 ab	9.55 ± 0.17 a	32.28 ± 0.76 ab	150.42 ± 4.00 bc
	S3	10.71 ± 0.57 a	1.14 ± 0.07 ab	9.55 ± 0.51 a	33.94 ± 0.87 ab	154.42 ± 4.00 ab
	S4	10.84 ± 0.48 a	1.16 ± 0.06 a	9.56 ± 0.56 a	34.54 ± 1.56 a	158.42 ± 4.00 a

Data are the mean ± SD. Different letters indicate statistical significance at the *p* < 0.05 level within the same column. S0, S1, S2, S3, and S4 are no straw return, one year of straw return, two successive years of straw return, three successive years of straw return, and four successive years of straw return, respectively.

**Table 3 plants-13-02446-t003:** Effects of different years of straw return on the CO_2_, CH_4_, and N_2_O emissions.

Year	Treatment	Cumulative CO_2_ Emissions (kg hm^−2^)	Cumulative CH_4_ Emissions (kg hm^−2^)	Cumulative N_2_O Emissions (kg hm^−2^)
2021	S0	12,061.23 ± 709.48 b	582.15 ± 30.76 b	0.86 ± 0.07 a
	S1	15,374.79 ± 1182.68 a	732.80 ± 56.62 a	0.80 ± 0.09 a
	S2	17,270.15 ± 1233.58 a	742.57 ± 40.42 a	0.87 ± 0.10 a
	S3	16,518.39 ± 1032.40 a	723.09 ± 59.06 a	0.81 ± 0.07 a
2022	S0	14,000.81 ± 823.58 b	545.80 ± 42.62 b	0.83 ± 0.08 a
	S1	15,656.78 ± 1204.37 ab	674.14 ± 31.00 a	0.76 ± 0.07 a
	S2	17,645.25 ± 980.29 a	704.79 ± 51.08 a	0.78 ± 0.09 a
	S3	16,862.02 ± 1053.88 a	667.43 ± 47.30 a	0.89 ± 0.09 a
	S4	17,437.49 ± 1162.50 a	649.67 ± 36.56 a	0.82 ± 0.07 a

Data are the mean ± SD. Different letters indicate statistical significance at the *p* < 0.05 level within the same column. S0, S1, S2, S3, and S4 are no straw return, one year of straw return, two successive years of straw return, three successive years of straw return, and four successive years of straw return, respectively.

## Data Availability

Data will be made available on request.

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
