# Peer review of "Successive Years of Rice Straw Return Increased the Rice Yield and Soil Nutrients While Decreasing the Greenhouse Gas Intensity"

_plants, 2024, doi:10.3390/plants13172446_

Round 1

Reviewer 1 Report

Comments and Suggestions for Authors

Comments and Suggestions for Authors

Title: Successive years of rice straw return increased rice yield and soil
nutrients, while decreasing the greenhouse gas intensity

Dear Authors

The scope of research results presented in the manuscript falls within the publishing profile of Plants journal. The research issues are important for soil protection in sustainable agriculture and environmental protection. The article is generally well written. The research results are presented clearly in tables and figures. The discussion of own results is well conducted in relation to other authors.

In order to increase the usefulness of the article, Authors must refer to the following points.

Remarks:

1)    Results: Figure 2E and 2F - It should be: nitrogen content in g kg-1 (Y axis). Figure 6 i 7 - The year of research should be added. Figure 9 - Please record the units correctly for nitrogen, carbon and dry matter. Subsection 2.6. - The description of CO2, CH4 and N2O emissions refers to Figure 11 (please correct). Subsection 2.8. - The correlation description refers to Figure 13 (please correct).

2)    Discussion: Adapt the citation policy in this section to publishing requirements. Line 501 - Is N2O an intermediate product in the nitrification process?

3)    Materials and Methods: Subsection 4.1. - When describing weather conditions, temperature ranges during the growing season should be given, not temperature sums. Soil type should be given according to the latest WRB (World Reference Base for Soil Resources) classification, 4th edition, 2022. Subsection 4.2. - Straw yields should be given in t ha-1. Please supplement potassium dose. Subsection 4.3.6 (Lines 653 - 654) The principle of total nitrogen determination should be improved.

Detailed remarks:

Line 33 – Rice, Rice ?????

Lines 69 – 70 –  References number must be added.

Line 94 – References number must be added.

Line 112 – it should be: Results

Line 551 – It should be: Figure 15.

Best regards

Author Response

Comments 1: Results: Figure 2E and 2F - It should be: nitrogen content in g kg-1 (Y axis). Figure 6 i 7 - The year of research should be added. Figure 9 - Please record the units correctly for nitrogen, carbon and dry matter. Subsection 2.6. - The description of CO2, CH4 and N2O emissions refers to Figure 11 (please correct). Subsection 2.8. - The correlation description refers to Figure 13 (please correct).

Response: We appreciate the reviewer’s kind suggestions. “nitrogen content in mg (Y axis)” was corrected as “nitrogen uptake per plant in mg(Y axis)”(Line 162, mark revisions in green). “Figure 6 and Figure 7” have be added the year of research (Line 229 and Line 242, mark revisions in green). The units of nitrogen, carbon, and dry matter in Figure 9 have been corrected (Line 290, mark revisions in green). “The description of CO2, CH4 and N2O emissions in Figure 11”have been corrected (Line 330, mark revisions in green). “The correlation description in Figure 13 have been corrected (Line 367, mark revisions in green).

Comments 2: Discussion: Adapt the citation policy in this section to publishing requirements. Line 501 - Is N2O an intermediate product in the nitrification process?

Response: We appreciate the reviewer’s kind suggestions. We have cited references for the statement in line 501 (Line 503, mark revisions in green).

Comments 3: Materials and Methods: Subsection 4.1. - When describing weather conditions, temperature ranges during the growing season should be given, not temperature sums. Soil type should be given according to the latest WRB (World Reference Base for Soil Resources) classification, 4th edition, 2022. Subsection 4.2. - Straw yields should be given in tha -1. Please supplement potassium dose. Subsection 4.3.6 (Lines 653 - 654) The principle of total nitrogen determination should be improved.

Response: Thanks for your useful comments. The weather conditions have been redescribed (Line 546-549, mark revisions in green). The soil type has been corrected according to WRB (Line 549-550, mark revisions in green). The straw yield has been added (Line 564, mark revisions in green). The amount of potassium fertilizer applied has been added (Line 570-571, mark revisions in green). The method for determining total nitrogen has been redescribed (Line 657-658, mark revisions in green).

Comments 4: Line 33 – Rice, Rice ?????

Response: Thanks for your useful comments. “Rice, Rice” was corrected as “Rice”(Line 33, mark revisions in green).

Comments 5: Lines 69 – 70 –  References number must be added.

Response: Thanks for your useful comments. References number have be added lines 69-70(Line 71, mark revisions in green).

Comments 6: Line 94 – References number must be added.

Response: Thanks for your useful comments. References number have be added line 94(Line 97, mark revisions in green).

Comments 7: Line 112 – it should be: Results

Response: Thanks for your useful comments. “Result” was corrected as “Results”(Line 112, mark revisions in green).

Comments 8: Line 551 – It should be: Figure 15

Response: Thanks for your useful comments. “Figure 14” was corrected as “Figure 15”(Line 554, mark revisions in green).

Reviewer 2 Report

Comments and Suggestions for Authors

Dear Authors,

I reviewed your manuscript on the effects of different straw return durations on soil fertility, rice nitrogen uptake, and greenhouse gas emissions. The study provides a comprehensive analysis of the effects of varying straw return durations on key aspects such as soil fertility, rice nitrogen uptake, and greenhouse gas emissions. The inclusion of multiple variables and detailed measurements enhances the robustness of your findings. The focus on practical aspects, including the impact of straw return on crop yield and soil health, is highly relevant for agricultural practices. Your work offers valuable insights into optimizing rice production and soil management. The data are presented effectively, with well-organized tables and figures that clearly illustrate the relationships between straw return duration, soil nutrients, crop yield, and greenhouse gas emissions.

However, I have some suggestions that could further enrich your research.

- The statistical methods used for analyzing greenhouse gas emissions and nutrient flows could be further detailed. Providing a more robust explanation of the statistical models and their assumptions would enhance the credibility of your results.

In the discussion section:

- A more detailed examination of the study's limitations would strengthen it. For example, addressing potential sources of error in greenhouse gas measurements and variations in soil conditions across different treatment plots would offer a more balanced perspective.

- While your study covers the immediate effects of straw return over several years, it would benefit from incorporating an extended analysis of the long-term sustainability of these practices. Including a discussion of the potential cumulative effects on soil health and crop productivity over multiple decades in a dedicated paragraph within the discussion section could provide a more comprehensive view of the long-term impact of straw return practices.

In addition to your extensive research, I recommend considering the following article for inclusion in your references: Deligios et al. (DOI: 10.1007/s13593-017-0465-3) which provides complementary insights into nutrient management and sustainable practices, particularly regarding long-term nutrient balance. It could offer additional perspectives that might complement your findings.

Author Response

Comments 1: - The statistical methods used for analyzing greenhouse gas emissions and nutrient flows could be further detailed. Providing a more robust explanation of the statistical models and their assumptions would enhance the credibility of your results.

Response: Thank you for pointing this out. We completely agree with the constructive comment.We have improved the statistical methods used to analyze greenhouse gas emissions and nutrient flow (Line 604-609, Line 656-661, Line 663-694, and Line 697-702, mark revisions in yellow).

Comments 2: In the discussion section: - A more detailed examination of the study's limitations would strengthen it. For example, addressing potential sources of error in greenhouse gas measurements and variations in soil conditions across different treatment plots would offer a more balanced perspective.

Response: Thanks for your useful comments. We will conduct a more detailed examination of the limitations of our research. Regarding the gas collection method, we followed the methods in the reference (Jiang et al. 2019) and (Tian et al. 2019).

Jiang, Y.; Qian, H.Y.; Huang, S.; Zhang, X.Y.; Wang, L.; Zhang, L.; Shen, M.X.; Xiao, X.P.; Chen, F.; Zhang, H.L.; Lu, C.Y.; Li, C.; Zhang, J.; Deng, A.X.; van Groenigen, K.J.; Zhang, W.J. Acclimation of methane emissions from rice paddy fields to straw ad-dition. Sci Adv. 2019, 5, eaau9038.

Tian, W.; Wu, Y.Z.; Tang, S.R.; Hu, Y.L.; Lai, Q.Q.; Wen, D.N.; Meng, L.; Wu, C.D. Effects of different fertilization modes on greenhouse gas emission characteristics of paddy fields in hot areas. Environmental Science. 2019, 40, 2426-2434.

Comments 3: - While your study covers the immediate effects of straw return over several years, it would benefit from incorporating an extended analysis of the long-term sustainability of these practices. Including a discussion of the potential cumulative effects on soil health and crop productivity over multiple decades in a dedicated paragraph within the discussion section could provide a more comprehensive view of the long-term impact of straw return practices.

Response: We completely agree with the constructive comment. We have supplemented the discussion section (Lines 405-410, Lines 445-447, Lines 496-499) with the potential cumulative impact of return straw on soil health and crop productivity over multiple decades (mark revisions in yellow).

Comments 4: In addition to your extensive research, I recommend considering the following article for inclusion in your references: Deligios et al. (DOI: 10.1007/s13593-017-0465-3) which provides complementary insights into nutrient management and sustainable practices, particularly regarding long-term nutrient balance. It could offer additional perspectives that might complement.

Response: Thanks for your useful comments. “Deligios et al. (DOI: 10.1007/s13593-017-0465-3)” has been supplemented (Line 748, mark revisions in yellow).